# Effect of *Lactiplantibacillus plantarum* on the Conversion of Linoleic Acid of Vegetable Oil to Conjugated Linoleic Acid, Lipolysis, and Sensory Properties of Cheddar Cheese

**DOI:** 10.3390/microorganisms11102613

**Published:** 2023-10-23

**Authors:** Awais Khan, Muhammad Nadeem, Fahad Al-Asmari, Muhammad Imran, Saadia Ambreen, Muhammad Abdul Rahim, Sadaf Oranab, Tuba Esatbeyoglu, Elena Bartkiene, João Miguel Rocha

**Affiliations:** 1Department of Dairy Technology, University of Veterinary and Animal Sciences, Lahore 54000, Pakistan; awaiskhan2483@gmail.com (A.K.); muhammad.nadeem@uvas.edu.pk (M.N.); 2Department of Food and Nutrition Sciences, College of Agricultural and Food Sciences, King Faisal University, Al-Ahsa 31982, Saudi Arabia; falasmari@kfu.edu.sa; 3Department of Food Science, Faculty of Life Sciences, Government College University, Faisalabad 38000, Pakistan; abdul.rahim@gcuf.edu.pk; 4University Institute of Food Science and Technology, The University of Lahore, Lahore 54590, Pakistan; saadia.ambreen@uifst.uol.edu.pk; 5Department of Food Science and Nutrition, Faculty of Medicine and Allied Health Sciences, Times Institute, Multan 60700, Pakistan; 6Department of Biochemistry, Faculty of Life Sciences, Government College University, Faisalabad 38000, Pakistan; oranab1022@gmail.com; 7Department of Food Development and Food Quality, Institute of Food Science and Human Nutrition, Gottfried Wilhelm Leibniz University Hannover, Am Kleinen Felde 30, 30167 Hannover, Germany; esatbeyoglu@lw.uni-hannover.de; 8Department of Food Safety and Quality, Veterinary Academy, Lithuanian University of Health Sciences, Tilzes Str. 18, LT-47181 Kaunas, Lithuania; elena.bartkiene@lsmuni.lt; 9Institute of Animal Rearing Technologies, Faculty of Animal Sciences, Lithuanian University of Health Sciences, Tilzes Str. 18, LT-47181 Kaunas, Lithuania; 10Universidade Católica Portuguesa, CBQF—Centro de Biotecnologia e Química Fina—Laboratório Associado, Escola Superior de Biotecnologia, Rua Diogo Botelho 1327, 4169-005 Porto, Portugal; 11LEPABE—Laboratory for Process Engineering, Environment, Biotechnology and Energy, Faculty of Engineering, University of Porto, Rua Dr. Roberto Frias, s/n, 4200-465 Porto, Portugal; 12ALiCE—Associate Laboratory in Chemical Engineering, Faculty of Engineering, University of Porto, Rua Dr. Roberto Frias, s/n, 4200-465 Porto, Portugal

**Keywords:** cheese, *Lp. plantarum*, conjugated linoleic acid, lipolysis

## Abstract

Conjugated linoleic acid (CLA) is perceived to protect the body from metabolic diseases. This study was conducted to determine the effect *of Lactiplantibacillus plantarum* (*Lp. plantarum*) on CLA production and sensory characteristics of cheddar cheese. *Lp. plantarum* can convert linoleic acid (LA) to CLA. To increase CLA in cheddar cheese and monitor the conversion of LA to CLA by *Lp. plantarum*, the LA content of cheese milk (3.4% fat) was increased by partially replacing fat with safflower oil (85% LA of oil) at 0, 3, 6, and 9% concentrations (T_1_, T_2_, T_3_, and T_4_). Furthermore, *Lp. plantarum* 10^8^ colony-forming units (CFU)/mL (8 log CFU mL^−1^) was added in all treatments along with traditional cheddar cheese culture (*Lactococcus lactis* ssp. *lactis* and *L. lactis* ssp. *cremoris*). After 30 days of ripening, *Lp. plantarum* in T_1_, T_2_, T_3_, and T_4_ was 6.75, 6.72, 6.65, and 6.55 log CFU g^−1^. After 60 days of ripening, *Lp. plantarum* in T_1_, T_2_, T_3_, and T_4_ was 6.35, 6.27, 6.19, and 6.32 log CFU g^−1^. After 60 days of ripening, *Lp. plantarum* in T_1_, T_2_, T_3_, and T_4_ was 6.41, 6.25, 6.69, and 6.65 log CFU g^−1^. GC-MS analysis showed that concentrations of CLA in the 90 days’ control, T_1_, T_2_, T_3_, and T_4_ were 1.18, 2.73, 4.44, 6.24, and 9.57 mg/100 g, respectively. HPLC analysis revealed that treatments containing *Lp. plantarum* and LA presented higher concentrations of organic acids than the control sample. The addition of safflower oil at all concentrations did not affect cheese composition, free fatty acids (FFA), and the peroxide value (POV) of cheddar cheese. Color flavor and texture scores of experimental cheeses were not different from the control cheese. It was concluded that *Lp. plantarum* and safflower oil can be used to increase CLA production in cheddar cheese.

## 1. Introduction

The globalization of functional foods has brought remarkable changes in dietary patterns, especially in developing countries. Changes in dietary patterns have been shown to be an important underlying factor for the increasing prevalence of chronic disorders and metabolic diseases. Therefore, increasing knowledge about the nutritional and health issues associated with the consumption of traditional foods has led researchers and consumers to become increasingly interested in the development of functional foods [1]. Conjugated linoleic acid (CLA)-supplemented foods have attracted great interest from consumers because of their anticarcinogenic, anti-atherogenic, antidiabetic, anti-inflammatory, and anti-obesity properties [2]. CLA occurs naturally in ruminant food products (beef, lamb, and dairy) due to bacterial biohydrogenation of linoleic acid in the rumen [3]. It is recommended to intake 2 g CLA/day, but the CLA content of dairy products is approximately 0.5 g/g of fat [2]. To achieve the nutritional need of CLA, a significant portion of dairy products and meat must be included in the daily diet, which may lead to excessive fat intake leading to several other health-related issues. It is possible to increase CLA in milk and dairy products by altering feeding practices; CLA in milk can be increased by adding vegetable oils, grazing on lush green pastures, and oilseeds. However, this is not a practical solution to the problem, since every farm has a different feeding strategy and dairy industries collect milk from thousands of farmers which is blended during the collection; the amount of CLA in milk is also affected by the season, with winter milk having more CLA than summer milk [4,5].

CLA can be synthetically produced from vegetable oils. In a recent study, vegetable oils were used as substrates to produce CLA from linoleic acid (LA) by urea treatment. It has been found that the change of LA to CLA is dependent on the concentration of LA. Safflower oil had more than 80% LA and yielded 31% CLA [6].

Conversion of linoleic acid to CLA through microbiological processes has been reported in the literature [7]; the addition of 0.02% LA to fermentation media considerably increased the production of CLA [8]. Among 14 genera of lactic acid bacteria (LAB), *Lactiplantibacillus plantarum* (*Lp. plantarum*) was found to be the highest converter of LA to CLA [9]. Certain strains of *Lb. acidophilus* converted LA to CLA [10]. Other researchers reported that certain strains of food fermenting bacteria, such as *Propionibacterium freudenreichii* ssp. *Freudenreichii*, *Propionibacterium freudenreichii* spp. *Shermanii*, and *Lp. plantarum*, can convert LA to CLA [11]. Strains of *Lp. plantarum* are widely used to accelerate the ripening process of certain varieties of cheese as they possess potent groups of enzymes to initiate several metabolic processes [12]. Guidone et al. [13] reported that LA can be converted to CLA by chemical and microbiological methods. However, the magnitude of conversion of LA to CLA by *Lp. plantarum* in cheese matrix has not been studied in detail.

According to the above-mentioned facts, it is evident that CLA production in food matrixes depends largely on the concentration of LA in the substrates. The LA content of cheddar cheese is approximately 2–2.5%. With this lower level of LA, it is practically impossible to raise CLA content by *Lp. plantarum* to the extent that can fulfill the recommended dietary allowance of CLA. In view of the current situation of metabolic diseases, it is technically challenging for food scientists to develop sustainable CLA-enriched foods using the mixed milk of hundreds from farms using different feed and managemental practices.

Safflower is produced in several countries of the world; it has already been used for a large number of food and medicinal applications. In comparison to all dietary lipids, it possesses the highest concentration of LA 85% fatty acid [14]. Safflower oil has a pale yellow color with an almost bland flavor and its suitability as a substrate for the conversion of LA to CLA in cheddar cheese needs to be explored. The literature supports the impression of using cheese as a potential vector for converting linoleic acid to CLA due to solid matrix, buffering capacity, and fat content. During the ripening of cheddar cheese, lactose metabolism, proteolysis, breakdown of lipids, production of free fatty acids (FFA), and flavoring compounds production [15]. The fatty acid composition of cheese before and after ripening was considerably different from each other [16]. CLA-enriched cheddar cheese may undergo lipid oxidation during the ripening phase; therefore, it is necessary to monitor lipid oxidation in CLA-enriched cheddar cheese. The use of safflower oil as a source of LA and *Lp. plantarum* as a microbiological converter of LA to CLA in cheese matrix has not been reported in the literature so far. The effect of four different concentrations of safflower oil and *Lp. plantarum* on the conversion of LA to CLA, fatty acids composition, organic acids, and lipolysis were determined with the latest analytical facilities.

## 2. Materials and Methods

### 2.1. Materials

Cheese starter culture, *L. lactis* ssp. *lactis* and *L. lactis* ssp. *cremoris* (DVS-RST-900), and *Lactiplantibacillus plantarum* “(*Lp. plantarum*)” (UALp-05) were purchased from Christian Hansen (Horsholm, Denmark). For analytical work, HPLC-grade chemicals were used (Sigma Aldrich, St. Louis, MO, USA). Safflower oil was obtained by mechanical expression of ‘20–25 °C; cold-pressed safflower oil was used for this study.

### 2.2. Cheddar Cheese Production and Experimental Plan

Milk (100 L for every treatment in replicate) from Holstein–Friesian cows having 3.4% fat and 3.2% protein content was pasteurized at 65 °C (30 min, batch pasteurization) and cooled to 31 °C. Cheese starter culture*, Lactococcus lactis ssp. lactis* and *Lactococcus lactis ssp. cremoris* (DVS culture @10 U/100 L milk), and *Lp. plantarum* and safflower oil were added as an experimental plan. Pre-acidification was conducted for 45 min (pH 6.4), and 17 mL calcium chloride (35%) and rennet 0.02% were mixed and left for 30 min for curd formation. Curd was then cut into 1.5–2 cm cubes, the temperature of the vat was increased to 39 °C in 45 min, and cooked at this temperature for 40 min. Whey was then drained, cheddaring was conducted till pH 5.2, followed by milling, salting (1.5%), pressing for 16 h (3.5 bar), and vacuum packaging. T1 comprised 100% milk fat and a (mixed/co-cultured) starter culture of *L. lactis* ssp. *lactis* and *L. lactis* ssp. c*remoris* and *Lp. plantarum* 10^8^ Cfu/mL. T2 comprised 97% milk fat, 3% safflower oil, and a starter culture of *L. lactis* ssp. *lactis* and *L. lactis* ssp. c*remoris* and *Lp. plantarum* 10^8^ Cfu/mL. T3 comprised 94% milk fat, 6% safflower oil, and a starter culture of *L. lactis* ssp. *lactis* and *L. lactis* ssp. c*remoris* and *Lp. plantarum* 10^8^ Cfu/mL. T4 comprised 91% milk fat, 9% safflower oil, and a starter culture of *L. lactis* ssp. *lactis* and *L. lactis* ssp. c*remoris* and *Lp. plantarum* 10^8^ Cfu/mL. The Control comprised 100% milk fat and a starter culture of *L. lactis* ssp. *lactis* and *L. lactis* ssp. c*remoris*. All kinds of cheeses were produced according to the standard protocol of cheddar cheese production and were ripened at 6–8 °C for the duration of 90 days and analyzed at pre-set intervals of 0, 45, and 90 days [14].

### 2.3. Analysis of Safflower Oil

Cold-extracted safflower oil was tested only for moisture, FFA, peroxide value (POV), iodine value, saponification value, and unsaponifiable matter by the standard protocols [17,18].

### 2.4. Milk and Cheese Composition

Fat, protein, solids-not-fat, total solids, pH, acidity, and mineral contents in cheese milk were determined by wet chemistry [19]. Moisture, fat, protein, and the pH of cheese were measured at 0, 45, and 90 days of ripening. Free fatty acids (FFA), color, moisture, peroxide value, refractive index, iodine value, and unsaponifiable matter were also tested [18].

#### 2.4.1. CLA Isomers and Other Fatty Acids

The fatty acid composition was analyzed at 0 and 90 days of this study on a GC-MS (7890-B, Agilent Technologies, Santa Clara, CA, USA) using an SP-2560 column (100 m × 0.25 mm id) and FID. For ester preparation, a 50 mg sample was reacted with 2 mL methanolic HCl in C_2_H_5_OH (15%) at 100 °C/60 min in a heating block, then cooled to 20–25 °C followed by the addition of 2 mL each *n*-hexane (99.99%) and deionized H_2_O. Test tubes were then mixed for 1 min; after 15 min, the upper layer was extracted and dried over Na_2_SO_4_ and put in GC-vials for injection (1 µL) by ALS at 1:50 split ratio. The temperature of the inlet and FID were 250 °C and He, O_2_, and H_2_ were flowing at 2, 4, and 40 mL/min. The total run time was 52 min. For comparison and quantification, FAME-37 (Supelco, St. Louis, MO, USA) standard was used, and for CLA isomers, SLB-IL111, and *cis*, the 11-*trans*-octadecadienoic acid solution was used [20].

#### 2.4.2. Population of *Lactiplantibacillus plantarum* (*Lp. plantarum*)

*Lp. plantarum* was determined after inoculation, pre-acidification, cooking, cheddaring, and 30, 60, and 90 days of maturation*. Lp. plantarum* was determined by the serial dilution of the sample in 0.85% solution of sodium chloride in MRS agar with vancomycin (20 mg/L, pH 5.6). Samples were incubated at 30 °C for 48 h, and colonies were counted and calculated as Cfu/mL. *Lp. plantarum* was determined after inoculation, pre-acidification, cooking, 156 cheddaring, and 30, 60, and 90 days of maturation [21].

#### 2.4.3. Organic Acids

In cheese samples, acetic acid (AA), butyric acid (BA), citric acid (CA), and lactic acid (LCA) were quantified by HPLC with Varian 9012 delivery system, auto-sampler, UV detector, Aminex-HPX-87H column (300 mm × 7.8 mm). Cheese samples of 5 g were blended with sulfuric acid (H_2_SO_4_) and nitric acid (HNO_3_, 25 mL, and 70 μL) and homogenized at 10,000 rpm. Furthermore, the fat layer was discarded and the central layer was extracted followed by heating at 50 °C/60 min, then centrifuged at 10,000× *g*, filtered, and transferred to HPLC vials. The mobile phase comprised 0.009 N H_2_SO_4_ and organic acids were measured at 220 nm [22].

#### 2.4.4. Lipolysis

Lipolysis in cheese samples was tested by analyzing the FFA, cholesterol, and peroxide value (POV) at 0, 45, and 90 days of refrigerated storage using standard methods [23].

### 2.5. Sensory Evaluation

Descriptive sensory analysis of CLA-enriched cheddar cheese samples was performed by 10 trained judges. For the pre-drafting of terminology (color, flavor, and texture), four sittings (each 2 h) were conducted. The color, flavor, and texture of cheddar cheese samples were analyzed in a well-ventilated and well-lit sensory evaluation laboratory at 20–25 °C using a nine-point scale and data were analyzed with FIZZ software (“FIZZ sensory software version 2.47B, Biosystèmes, Courtenon, France”). For palate cleansing, unsalted crisp bread was provided in individual sensory evaluation booths [24]. For sensorial testing, all applicable regulations were followed and informed consent was signed by all participants in the test.

### 2.6. Statistical Analysis

The experiment was designed in a completely randomized design and data were analyzed by two-way ANOVA to determine the impact of treatments. For the determination of significant differences among the means, Duncan’s multiple range (DMR) was applied using SAS 9.4 software at a 0.05 level of significance.

## 3. Results and Discussion

Free fatty acids (FFA) (0.14%), moisture content (0.19%), peroxide value (POV) (0.25 MeqO_2_/kg), iodine value (141.5 cg/100 g), unsaponifiable matter (1.34%), saponification value (191.7 mg KOH/g), and refractive index (1.4745) were found in cold extracted safflower oil. Fat, protein, lactose, mineral, solids-not-fat, total solids contents, pH, and acidity in cow milk were 3.4%, 3.21%, 4.63%, 0.72%, 8.61, 11.96%, 6.67, and 0.14% (lactic acid). The results of the chemical characteristics of safflower oil are not different from the literature of Milesi et al. [25]. The addition of *Lactiplantibacillus plantarum* (*Lp. plantarum)* and safflower oil from 3 to 9% concentrations did not affect cheddar cheese composition (Table 1). Cheddar cheese was prepared using *Lp. plantarum* as an adjunct starter culture, with fat, protein, and moisture contents similar to standard cheddar cheese [12]. In another similar research work, starter cultures did not affect the chemical composition of all dry-salted cheeses over 4 months of ripening [26]*. Lp. plantarum* did not affect the chemical composition of cheddar cheeses [27].

Murtaza et al. reported that the addition of adjunct starter cultures did not affect the moisture, fat, protein, and pH of cheese [28]. The chemical composition of cheddar cheese prepared from *Lactobacillus acidophilus* and *Bifidobacterium bifidum* cultures was not different from standard cheese. In a relevant study, chia oil was added to cheddar cheese in 2.5–10% concentrations, and fat, protein, and moisture contents were similar to the standard cheddar cheese (*p* > 0.05) [29]. Ahmed et al. [30] prepared Gouda cheese with mango kernel oil, and researchers reported that up to a 10% level, the composition of Gouda cheese was not influenced. In a similar investigation, when low melting point fractions of milk were transformed to cheddar cheese, it was found that the addition of low melting point fractions did not alter the chemical composition of cheddar cheese [31].

### 3.1. Population of Lactiplantibacillus plantarum (Lp. plantarum)

The population of *Lp. plantarum* was determined at different stages of cheese production and ripening (Figure 1). *Lp. plantarum* was not affected by the three different concentrations of safflower oil, the processes of cheese production, and the ripening period. The results of the population of *Lp. plantarum* are highly significant from technological and industrial adaptation viewpoints, as no additional technique for safeguarding the bacterial strain is required. Moreover, cheese industries can easily adopt the technology of CLA-enriched cheddar cheese, as no alteration in the cheese production process and additional processing or packaging equipment is required. The population of *Lp. plantarum* in cheddar cheese has been reported; however, its population in cheese in the presence of safflower oil (as a source of linoleic acid) has not been previously studied.

In a previous investigation, probiotic cheddar cheese was produced using *Lp. plantarum* in combination with traditional starter culture, and *Lp. plantarum* survived the cheese production and ripening phase with no decline in population till 12 weeks of storage [26]. In another study, the populations of *L. acidophilus* and *Lactobacillus* increased during the ripening period [32]. In a previous study, the populations of *Bifidobacterium lactis* B 94 and *L. casei* 01 were analyzed in cheese during the 60 days of ripening. In the first 15–30 days, the populations of both bacteria increased by two log cycles; however, their populations remained almost constant till 60 days of ripening [33]. Cheddar cheese was enriched with *Bifidobacterium* and *Lactobacillus* and the bacterial population was monitored till 10 weeks of ripening. It was found that the populations of both strains ranged from 10^6^ to 10^7^/g after the end of the ripening phase [34].

### 3.2. Fatty Acid Composition

In the current investigation effect of *Lactiplantibacillus plantarum* (*Lp. plantarum)*, it was determined on the conversion of LA to CLA using safflower oil as substrate (Table 2). It was recorded that *Lp. plantarum* efficiently converted LA of milk (T_1_) and safflower oil (T_2_, T_3_, and T_4_) to CLA isomers. However, conversion was largely dependent upon the availability of LA in the reaction substrate. The LA content of safflower oil was 85.19%, which was the main reason for the rise of CLA content in T_2_, T_3_, and T_4_. During the ripening of 90 days, concentrations of CLA in all the treatments and the control significantly increased; however, the rise in CLA was dependent upon the extent of LA in the substrate, and treatments having more linoleic acid underwent more CLA production.

GC-MS analysis showed that after the end of ripening (90-Days), the content of CLA in the control, T_1_, T_2_, T_3_, and T_4_ was 1.46, 2.18, 4.44, 6.24, and 9.75 mg/100 g (*p* < 0.05), respectively. Due to the conversion of LA to CLA during the ripening phase, concentrations of LA in 90-Days T_1_, T_2_, T_3_, and T_4_ were considerably lower than the values recorded at 0-Day (*p* < 0.05). At the end of ripening, concentrations of LA in the control, T_1_, T_2_, T_3_, and T_4_ were 1.87, 2.18, 1.34, 0.98, and 1.11 mg/100 g only, respectively. Before ripening, the LA content of the control, T_1_, T_2_, T_3_, and T_4_ were 2.56, 5.52, 5.98, 8.25, and 11.49 mg/100 g, respectively. Conversion of linoleic acid to CLA was largely dependent upon the concentration of linoleic acid in the substrate. The fatty acid profile of fermented milk and probiotic cheese was used as an indication of lipolysis and flavor production [35,36]. In the control (at 0-Day), concentrations of ∆9*c*,11*t*−18:2, ∆10*t*,12*c*−18:2, ∆9*c*,11*c*−18:2, ∆9*t*, 11*c*−18:2, ∆10*c*,12*t*−18:2, ∆8,9,11,10,12*c*−c18:2, and ∆8,9,11,10,12*t*−t18:2 were 0.23, 0.10, 0.09, 0.05, 0.08, 0.03, and 0.04 mg/100g, respectively. In the control (after 90-Days), concentrations of ∆9*c*,11*t*−18:2, ∆10*t*,12*c*−18:2, ∆9*c*,11*c*−18:2, ∆9*t*, 11*c*−18:2, ∆10*c*,12*t*−18:2, ∆8,9,11,10,12*c*−*c*18:2, and ∆8,9,11,10,12*t*−*t*18:2 were 0.37, 0.16, 0.14, 0.09, 0.17, 0.05, and 0.07 mg/100g, respectively. In T_4_ (at 0-Day), concentrations of ∆9*c*,11*t*−18:2, ∆10*t*,12*c*−18:2, ∆9*c*,11*c*−18:2, ∆9*t*, 11*c*−18:2, ∆10*c*,12*t*−18:2, ∆8,9,11,10,12*c*−*c*18:2, and ∆8,9,11,10,12*t*−*t*18:2 were 0.26, 0.12, 0.10, 0.15, 0.16, 0.09, and 0.13 mg/100g, respectively. In T_4_ (after 90-Days), concentrations of ∆9*c*,11*t*−18:2, ∆10*t*,12*c*−18:2, ∆9*c*,11*c*−18:2, ∆9*t*, 11*c*−18:2, ∆10*c*,12*t*−18:2, ∆8,9,11,10,12*c*−*c*18:2, and ∆8,9,11,10,12*t*−*t*18:2 were 1.91, 1.54, 1.48, 1.39, 1.18, 1.10, and 0.82 mg/100 g, respectively.

Cheese produced using *Bifidobacterium lactis* and *Lactobacillus casei* was analyzed for CLA content for 60 days of ripening, and after 15 days, cis-9, *trans*-11-C18:2, CLA; *trans*-10, cis-12-C18:2, CLA; and *trans*-9, *trans*-12- C18:2, CLA were found, and all the determination intervals till 60 days of ripening showed an increasing trend (1.8–7.9 times higher) [37]. Phillips et al. [33] produced cheese by blending cow and buffalo milk using *Lactobacillus casei* and *Lactobacillus acidophilus* to adjudge the production of CLA during the ripening phase of 90 days. The results revealed that concentrations of CLA increased during the whole ripening phase, and the rise in CLA was due to the conversion of LA to CLA. The literature suggests that certain bacteria have the ability to efficiently produce CLA from LA. In the current investigation, it was established that *Lp. plantarum* has the ability to convert LA to CLA. Ahmed et al. [30] reported that increasing LA in Gouda cheese considerably increased the production of CLA in mature cheese.

### 3.3. Organic Acids

The production of organic acids in cheese ripening has driven a considerable volume of research, which has found that the sensorial and nutritional characteristics of ripened cheese largely depend upon lipolysis, and the metabolic activities of bacterial cultures are also estimated by the extent of organic acids produced [38].

In the current study, concentrations of BA, CA, AA, and LCA in the control and experimental samples increased during the entire ripening phase of 90 days but to different extents (Figure 2). During cheese ripening, the production of organic acids was dependent upon the concentration of LCA present in cheese matrix and *Lactiplantibacillus plantarum* (*Lp. plantarum}*. The control sample had 2.56% LA of milk fat own origin with no addition of *Lp. plantarum*, and analysis intervals of 0, 45, and 90 days revealed significantly lesser values of BA, CA, AA, and LCA than all experimental samples. At the end of the ripening phase (90 days), the concentrations of BA, CA, AA, and LA in the control were 1076, 494, 417, and 16,118 ppm, respectively (*p* < 0.05). T_1_ showed significantly higher concentrations of BA, CA, AA, and LCA than the control at all three determination intervals. T_1_ had 2.52% LCA and 10^8^/g *Lp. plantarum*, and the concentrations of BA, CA, AA, and LA after 90 days of ripening were 1472, 1220, 481, and 16,891 ppm, respectively. A similar trend was observed after 60 days of analysis of organic acids. HPLC analysis showed that treatments that had higher extents of LA in the presence of *Lp. plantarum* yielded more BA, CA, AA, and LA (*p* < 0.05) after 45 and 90 days of the ripening phase. After 90 days of ripening, the amounts of BA in T_2_, T_3_, and T_4_ were 1792, 2875, and 3015 ppm (*p* < 0.05), respectively. At the end of the ripening phase (90 days), the amounts of CA in T_2_, T_3_, and T_4_ were 1475, 1579, and 1743 ppm (*p* < 0.05), respectively. At the end of the ripening phase (90 days), the amounts of AA in T_2_, T_3_, and T_4_ were 537, 849, and 1375 ppm (*p* < 0.05), respectively.

*Lp. plantarum* is capable of producing organic acids in LA-enriched cheddar cheese. *Lp. plantarum* is a technologically significant lactic acid bacterium that is highly prevalent in fermented foods of vegetable and animal origins, and it can tolerate low pH values and relatively high NaCl contents [39]. In cheese, acetic acid contributes to the development of flavor, and microbial activities lead to the production of acetic acid from lactose and citrate. Heterofermentative lactobacilli can also produce citric acid from lactose and amino acid [40]. Bacterial activities, fat hydrolysis, and added acids may lead to the generation of organic acids throughout ripening. A mechanism for the generation of organic acids in dairy products has been defined at a symposium on flavor in dairy foods and meat: oxidation converts LA to hydroperoxides which are transformed into ketones, aldehydes, and organic acids [37]. Ullah et al. [29] produced cheddar cheese using traditional cheddar, chia oil, and milk fat blends; the effect was estimated on the production of organic acids till 120 days. From the results, it was concluded that cheese samples having chia oil had higher amounts of organic acids than cheese produced from 100% milk fat.

### 3.4. Lipolysis

In cheese matrix, the hydrolysis of fat can take place due to several reasons, such as the presence of indigenous and bacterial lipases, moisture contents, metal ions, etc., and it leads to the liberation of fatty acids from triglyceride molecules—and these are then referred to as FFA. In almost all ripened cheeses, hydrolysis takes place; however, the degree of lipolysis may vary from one variety of cheese to another. In certain hard Italian varieties, excessive lipolysis takes place and higher amounts of FFA are produced [41]. FFAs play an imperative part in the expansion of flavors in ripened cheese. In this research study, the FFA content of the freshly prepared control and CLA-enriched cheddar cheese was not different (*p* > 0.05) (Table 3). At the end of ripening, the highest estimated FFA content was in T_4_ followed by T_3_ and T_2_, which indicated that *Lactiplantibacillus plantarum* (*Lp. plantarum)* did not play any significant role in the hydrolysis of fat molecules. The production of FFA was mainly dependent upon the ripening period and the presence of safflower oil from 3 to 9% (used as a source of LA). The ripening period and concentration of safflower oil played their roles in the production of FFA: 90-day-old samples had more FFA than 45-day-old samples. Similarly, cheese samples having more safflower oil generated more FFA during the ripening period of 90 days. Higher FFA in cheddar cheese containing vegetable oil is supported by the literature.

In the present study, a change in the amount of cholesterol was used as an indication of lipolysis. The addition of safflower oil and ripening both affected the cholesterol content of cheese samples. At 0-Days, the lowest cholesterol was found in T_4_, followed by T_3_ and T_2_, while T_1_ and the control almost had the same level of cholesterol. During ripening, a decline in cholesterol was dependent upon the ripening period, and non-significant differences in the cholesterol content of the control and T_1_ showed that *Lp. plantarum* did not affect cholesterol concentration during the ripening phase of cheese.

Dairy products enriched with unsaturated fatty acids are more exposed to lipid oxidation. Therefore, in this investigation, the POV of all types of cheese samples were analyzed during the whole ripening period at pre-set intervals. Several methods of lipid oxidation have been discovered, such as the induction period by Rancimat; however, accelerated methods of lipid oxidation are not traditionally used for dairy products as these methods were established to determine the oxidative stability of plant-based oils. The peroxide value of the control and CLA-enriched cheddar samples increased in the ripening phase. The POV of all types of cheese samples was within the permissible limits of the EU. Oxidative stability (induction period) of CLA isomers was in the order of *t,t*-CLA > *c,t*-CLA > *c,c*-CLA [12].

### 3.5. Sensory Evaluation

The color, flavor, and texture of the control, T_1_, T_2_, T_3_, and T_4_ were not significantly different from each other. *Lactiplantibacillus plantarum (Lp. plantarum)* and safflower oil from 3 to 9% concentrations did not affect the sensory properties of cheese samples (Figure 3). After 90 days of ripening, the flavor scores of the control, T_1_, T_2_, T_3_, and T_4_ were 91, 90, 88, 91, and 92% of the total score (9), respectively. Sensory observation is a straightforward parameter to judge customer likes and dislikes. 

## 4. Conclusions

The addition of *Lactiplantibacillus plantarum* and safflower oil from 3 to 9% concentrations did not affect cheddar cheese composition. *Lactiplantibacillus plantarum* efficiently converted LA to CLA isomers. In T_3_ and T_4_ (6 and 9% safflower, respectively, and starter culture of *L. lactis* ssp. *Lactis* (*L. lactis* ssp. *Lactis*) and *L. lactis* ssp. C*remoris* and *Lp. plantarum* 10^8^/^mL^), CLA was 6.24 and 9.75 mg/100 g. To prevent metabolic disease, CLA should be taken on a regular basis. The peroxide value of CLA-enriched cheese was within the allowable limits of EU. Overall, *Lp. plantarum* and safflower oil from 6 to 9% concentrations can be used to produce CLA-enriched cheddar cheese. Further research work should be conducted on the application of *Lactiplantibacillus plantarum* in different foods.

## Figures and Tables

**Figure 1 microorganisms-11-02613-f001:**
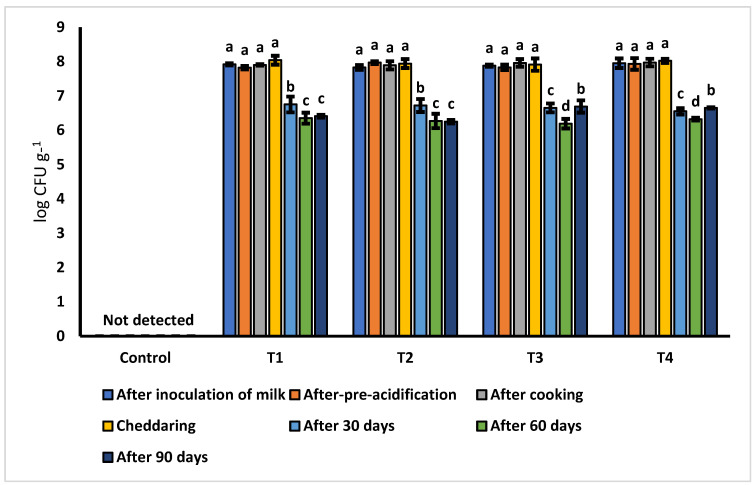
Survival of *Lp. plantarum* (log CFU g^−1^) in ripening phase.

**Figure 2 microorganisms-11-02613-f002:**
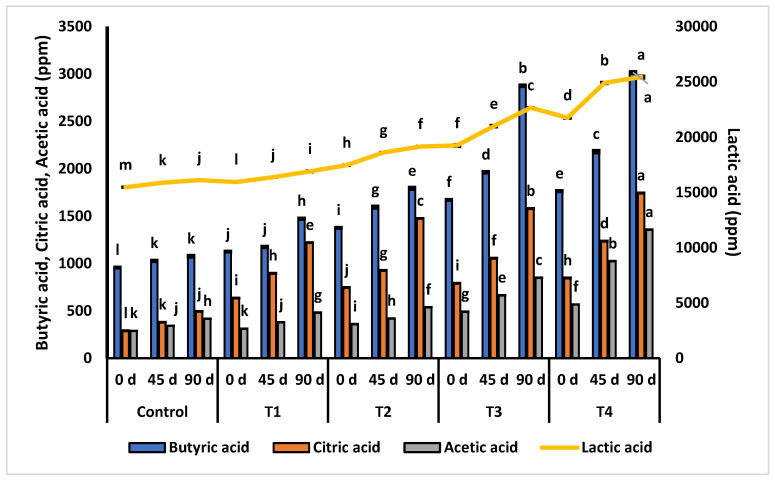
Effect of *Lp. plantarum* and safflower oil on the organic acids of cheddar cheese.

**Figure 3 microorganisms-11-02613-f003:**

Effect of *Lp. plantarum* and safflower oil on sensory characteristics (color, flavor, texture) of cheddar cheese.

**Table 1 microorganisms-11-02613-t001:** Composition of cheddar cheese.

Attribute	Control	T_1_	T_2_	T_3_	T_4_
Moisture%	40.52 ± 1.03 ^a^	41.22 ± 1.05 ^a^	40.16 ± 1.02 ^a^	41.11 ± 1.06 ^a^	40.85 ± 1.02 ^a^
Fat%	30.47 ± 0.9 ^a^	29.91 ± 0.9 ^a^	30.17 ± 0.9 ^a^	30.37 ± 0.9 ^a^	29.78 ± 0.9 ^a^
Protein%	26.19 ± 0.7 ^a^	26.37 ± 0.7 ^a^	26.14 ± 0.7 ^a^	26.65 ± 0.7 ^a^	26.88 ± 0.7 ^a^
pH	5.24 ± 0.2 ^a^	5.22 ± 0.2 ^a^	5.21 ± 0.2 ^a^	5.23 ± 0.2 ^a^	5.22 ± 0.2 ^a^

^a^ All means mentioned in the rows of moisture, fat, protein, and pH showed a non-significant (*p* > 0.05) variation.

**Table 2 microorganisms-11-02613-t002:** Fatty acids composition of cheddar cheese added with *Lp. plantarum* (mg/100 g).

Fatty Acid	Safflower Oil	Control		T_1_		T_2_		T_3_		T_4_	
		0-Day	90-Days	0-Day	90-Days	0-Day	90-Days	0-Day	90-Days	0-Day	90-Days
C4:0	Not Found	1.89 ± 0.01 a	1.90 ± 0.01 a	1.85 ± 0.02 a	1.80 ± 0.01 a	1.79 ± 0.03 a	1.75 ± 0.02 a	1.77 ± 0.01 a	1.70 ± 0.02 b	1.78 ± 0.02 a	1.72 ± 0.01 b
C6:0	Not Found	2.36 ± 0.03 a	2.32 ± 0.06 a	2.34 ± 0.04 a	2.28 ± 0.05 a	2.26 ± 0.01 a	2.23 ± 0.01 a	2.25 ± 0.04 a	2.13 ± 0.01 a	2.21 ± 0.01 a	2.10 ± 0.01 b
C8:0	Not Found	2.51 ± 0.06 a	2.49 ± 0.05 a	2.48 ± 0.03 a	2.42 ± 0.07 a	2.40 ± 0.04 a	2.25 ± 0.03 b	2.37 ± 0.02 a	2.23 ± 0.02 b	2.31 ± 0.03 a	2.18 ± 0.06 b
C10:0	Not Found	2.77 ± 0.10 a	2.74 ± 0.10 a	2.75 ± 0.07 a	2.70 ± 0.09 a	2.67 ± 0.02 a	2.62 ± 0.02 b	2.75 ± 0.04 a	2.61 ± 0.07 b	2.70 ± 0.04 a	2.57 ± 0.02 c
C12:0	Not Found	2.94 ± 0.12 a	2.93 ± 0.13 a	2.93 ± 0.11 a	2.87 ± 0.02 a	2.86 ± 0.22 a	2.74 ± 0.06 b	2.83 ± 0.06 a	2.70 ± 0.08 b	2.77 ± 0.03 a	2.53 ± 0.07 c
C14:0	0.11 ± 0.01 f	11.25 ± 0.23 a	11.13 ± 0.02 a	10.65 ± 0.16 b	10.49 ± 0.24 b	10.33 ± 0.17 b	9.98 ± 0.42 b	9.14 ± 0.16 c	8.55 ± 0.02 d	8.51 ± 0.20 d	7.66 ± 0.11 e
C16:0	1.22 ± 0.02 h	26.74 ± 0.29 a	26.25 ± 0.74 a	25.16 ± 0.33 b	24.75 ± 0.45 c	24.37 ± 0.31 c	23.38 ± 0.51 d	22.74 ± 0.31 e	21.27 ± 0.52 f	21.20 ± 0.08 f	18.41 ± 0.28 g
C18:0	4.74 ± 0.15 f	8.19 ± 0.15 a	8.13 ± 0.20 a	7.81 ± 0.17 b	7.44 ± 0.35 b	7.16 ± 0.10 c	6.62 ± 0.09 d	6.11 ± 0.09 d	5.43 ± 0.19 e	5.35 ± 0.13 e	4.11 ± 0.16 g
C18:1	6.89 ± 0.13 h	23.94 ± 0.34 a	21.17 ± 0.16 c	23.51 ± 0.54 a	22.98 ± 0.77 b	22.36 ± 0.29 b	21.87 ± 0.39 c	20.19 ± 0.37 d	19.13 ± 0.22 e	18.98 ± 0.09 f	16.14 ± 0.18 g
C18:2	85.19 ± 0.22 a	2.56 ± 0.21 e	1.87 ± 0.16 g	2.52 ± 0.09 e	2.18 ± 0.04 f	5.98 ± 0.05 d	1.34 ± 0.42 h	8.25 ± 0.24 c	0.98 ± 0.17 j	11.49 ± 0.12 b	1.11 ± 0.02 i
∆9*c*,11*t*-18:2	Not Found	0.23 ± 0.09 f	0.37 ± 0.02 e	0.22 ± 0.03 f	0.58 ± 0.10 d	0.22 ± 0.02 f	1.12 ± 0.03 c	0.21 ± 0.01 f	1.34 ± 0.04 b	0.26 ± 0.04 f	1.91 ± 0.03 a
∆10*t*,12*c*-18:2	Not Found	0.10 ± 0.06 f	0.16 ± 0.01 e	0.11 ± 0.01 f	0.42 ± 0.02 d	0.08 ± 0.01 f	0.79 ± 0.15 c	0.07 ± 0.01 f	1.12 ± 0.02 b	0.12 ± 0.02 f	1.54 ± 0.06 a
∆9*c*,11*c*-18:2	Not Found	0.09 ± 0.02 f	0.14 ± 0.03 e	0.08 ± 0.01 f	0.47 ± 0.05 d	0.07 ± 0.01 f	0.91 ± 0.01 c	0.09 ± 0.02 f	1.05 ± 0.01 b	0.10 ± 0.03 f	1.48 ± 0.02 a
∆9t, 11c-18:2	Not Found	0.05 ± 0.01 f	0.09 ± 0.01 e	0.06 ± 0.01 f	0.29 ± 0.06 d	0.06 ± 0.01 f	0.48 ± 0.05 c	0.11 ± 0.01 e	0.92 ± 0.02 b	0.15 ± 0.01 e	1.39 ± 0.1 a
∆10c,12*t*-18:2	Not Found	0.08 ± 0.02 f	0.17 ± 0.01 e	0.19 ± 0.02 e	0.31 ± 0.02 d	0.05 ± 0.01 g	0.35 ± 0.03 c	0.09 ± 0.01 f	0.68 ± 0.03 b	0.10 ± 0.05 f	1.18 ± 0.01 a
∆8,9,11,10,12c-c18:2	Not Found	0.03 ± 0.01 f	0.05 ± 0.02 f	0.06 ± 0.02 f	0.15 ± 0.01 d	0.04 ± 0.01 f	0.27 ± 0.02 c	0.05 ± 0.01	0.51 ± 0.01 b	0.09 ± 0.06 e	1.10 ± 0.02 a
∆8,9,11,10,12t-t18:2	Not Found	0.04 ± 0.01 f	0.07 ± 0.01 e	0.08 ± 0.01 e	0.12 ± 0.01 d	0.05 ± 0.01 f	0.21 ± 0.01 c	0.07 ± 0.02 e	0.38 ± 0.01 b	0.13 ± 0.02 d	0.82 ± 0.04 a
C18:3	0.25 ± 0.02	0.58 ± 0.02 a	0.41 ± 0.01 c	0.56 ± 0.1 a	0.39 ± 0.03 c	0.53 ± 0.03 a	0.31 ± 0.02 d	0.49 ± 0.04 b	0.24 ± 0.02 e	0.25 ± 0.01 e	0.15 ± 0.01 f
∑ CLA	-	1.18 ± 0.02 f	1.46 ± 0.04 e	1.36 ± 0.09 e	2.73 ± 0.03 d	1.10 ± 0.01 f	4.44 ± 0.11 c	1.18 ± 0.06 f	6.24 ± 0.15 b	1.20 ± 0.03 f	9.57 ± 0.19 a

In rows of this table, if means are bearing a different letter, it reveals a statistically significant variation (*p* < 0.05).

**Table 3 microorganisms-11-02613-t003:** Effect of *Lp. plantarum* and safflower oil on lipolysis of cheddar cheese.

Treatments	Ripening Days	FFA%	Cholesterol (mg/100 g)	POV (MeqO_2_/kg)
Control	0	0.08 ± 0.01 ^h^	165 ± 1.27 ^a^	0.24 ± 0.02 ^e^
	45	0.13 ± 0.02 ^g^	155 ± 1.14 ^b^	0.25 ± 0.01 ^e^
	90	0.17 ± 0.01 ^f^	121 ± 0.87 ^c^	0.45 ± 0.05 ^d^
T_1_	0	0.08 ± 0.02 ^h^	161 ± 1.26 ^a^	0.25 ± 0.03 ^e^
	45	0.14 ± 0.03 ^f^	144 ± 1.49 ^b^	0.28 ± 0.02 ^e^
	90	0.21 ± 0.01 ^d^	119 ± 0.45 ^c^	0.47 ± 0.06 ^d^
T_2_	0	0.09 ± 0.01 ^h^	158 ± 0.88 ^b^	0.22 ± 0.02 ^e^
	45	0.19 ± 0.04 ^e^	139 ± 0.78 ^d^	0.30 ± 0.05 ^e^
	90	0.25 ± 0.02 ^c^	122 ± 0.26 ^e^	0.64 ± 0.07 ^c^
T_3_	0	0.08 ± 0.01 ^h^	155 ± 0.96 ^b^	0.26 ± 0.04 ^e^
	45	0.22 ± 0.01 ^d^	125 ± 1.32 ^e^	0.29 ± 0.02 ^e^
	90	0.29 ± 0.02 ^b^	114 ± 0.66 ^f^	0.72 ± 0.01 ^b^
T_4_	0	0.08 ± 0.01 ^h^	152 ± 1.19 ^b^	0.23 ± 0.01 ^e^
	45	0.26 ± 0.02 ^c^	121 ± 1.55 ^e^	0.31 ± 0.02 ^e^
	90	0.37 ± 0.01 ^a^	78 ± 1.31 ^g^	0.85 ± 0.03 ^a^

In columns of this table, if means are bearing a different letter, it reveals a statistically significant variation (*p* < 0.05).

## Data Availability

No new data were created or analyzed in this study. Data sharing is not applicable to this article.

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
