# Peer review of "Effect of Lactiplantibacillus plantarum on the Conversion of Linoleic Acid of Vegetable Oil to Conjugated Linoleic Acid, Lipolysis, and Sensory Properties of Cheddar Cheese"

_microorganisms, 2023, doi:10.3390/microorganisms11102613_

Round 1

Reviewer 1 Report (Previous Reviewer 2)

The authors have thoroughly revised their manuscript, providing an improved version that responded satisfactorily to the questions raised in the previous round of revision.

One very minor remark: A final stop is missing on line 192.

Also, the statment of ethical compliance, which the authors show on line 192, must be given at the end of the manuscript, right after the "Authors contributions" section and, as required by MDPI, it should be in the lines of “All subjects gave their informed consent for inclusion before they participated in the study. The study was conducted in accordance with the Declaration of Helsinki, and the protocol was approved by the Ethics Committee XXXX” (Ethics Committee of the author’s institution, for instance) (https://www.mdpi.com/ethics).

Author Response

Reviewer 2 Report (New Reviewer)

“Effect of Lactiplantibacillus plantarum on the Conversion of 2 Linoleic Acid of Vegetable Oil to Conjugated Linoleic Acid, Li-3 polysis and Sensory Properties of Cheddar Cheese”

The authors investigated the use of safflower oil as a source of LA and Lp. plantarum as a microbiological converter of LA to CLA in cheese matrix. The effect of four different concentrations of safflower oil and Lp. plantarum on CLA, organic acids and lipolysis was evaluated.

There were many observations to be made, and they are reported as follows:

English should be improved.

It would be necessary for the authors to check the content of the articles cited in the text.

The abstract does not specify the aim of the study. From line 50 to line 57: the abbreviations can be written in the text

Lines 66- 67 : It would be appropriate to include a reference related to the specific indications reported by the authors.

Line 77: “LA”; it would be better to write “Linoleic acid (LA)”

Line 81:”…; addition of 0.02%....”refers to other authors. It would be better to write “.Addition of 0.02%....”

Line 85: the reference [9] should be reported in line 87, after “….can convert LA to CLA [9]”

Line 88: “plantarum” (lowercase-Italics)

Line 89: reference [10] refers to “E. Rosberg-Cody et al., 2004”; these authors do not speak about L. plantarum. The authors should change the reference.

Lines 90-91: the authors write “Nadeem et al. [11] reported that LA can be converted to CLA by chemical and microbiological methods. However, magnitude of conversion…..”. Where in this article are these statements mentioned? Furthermore, this article does not mention Lp.plantarum and the food matrix is margarine, not cheese

Lines 97-97: “…..one slice of cheese (30 g approximately)…..CLA”. A reference should be inserted after this sentence

Lines 97-98: “In view of the current…… CLA enriched foods”. This sentence is not clear to the reader; there should be a more clearly worded statement.

Line 102: “……LA 85% fatty acid [12]”. Article n.12 does not mention safflower. (Batool et al. Impact of vitamin E and selenium on antioxidant capacity and lipid oxidation of cheddar cheese in accelerated ripening. Lipids Health Dis. 2018, 17, 1-14

Lines 106-107: Article n.13 does not mention “….lactose metabolism, proteolysis…..”.

Lines 108-109: the sentence “Fatty acid composition of cheese before and after ripening  were considerably different from each other “ is related to the reference  [14]. Kosikowski, F.J.C.; foods, f.m. Cheddar cheese and related types. 1982, 204-225”. This reference is not written correctly; this does not allow the reader to check the content of the article. Authors should report the reference correctly.

Lines 113-114: except for Lp.plantarum, the sentence should not be written in Italics

Lines 113-115: It is advisable that the aim of the study should be more clearly stated.

Line 119: “(Lp. plantarum)

Lines 119-120: it should be better to write “Chr. Hansen (Hørsholm,Denmark)”

Line 120: screwpress”. It would be appropriate for the authors to include some details of the press used and the name and nationality of the manufacturer.

Line 122: “20oC”

Line 123: “2.2. Cheddar Cheese Production and Experimental Plan”. This paragraph should be rewritten in a logical order. In particular: 1) describe the milk; 2) pasteurization process and cooling; 3) description of the concentrations of safflower oil as a fat substitute in the milk cheese, as stated in the abstract [i.e. safflower oil (85% LA of oil) at 0, 3, 6 and 9% concentrations (T1, T2, T3 and T4); 4) cheesemaking and vacuum packaging; 5) ripening; 6) samples analyses

It would be appropriate to specify how the percentages of fat and total protein have been determined and how the milk has been pasteurised.

Line 132: the authors report the reference [15] “AOAC. Official Methods of Analysis of AOAC International. 18th Edition, AOAC International, Gaitherburg. 2011. 2590”. This reference is written after the cheese-making process, but it is not clear what analyses were carried out. The sentence needs to be rewritten more clearly.

Lines 134- 135: “…108 CFU/ml….”

Lines 133,134,135,136, 137,138 139, 140, 141, 142 and 166: the numbers 125,126,127,128, 129, 130, 131,132, 133 134, and 156 are erroneously given; they must be deleted. Also, in the References section, a lot of irrelevant numbers in the text need to be deleted.

Lines 139-140: for the Control, the amount of starter (L. lactis ssp. lactis and L. lactis ssp. cremoris) must be reported.

Line140: ”All kinds of cheeses were produced according to the standard  protocol of cheddar cheese”. At the end of this sentence there should be a reference

Line 141: “…6-8oC…”

Line 145: “…by the standard protocols [16]”. It would be better to write “…through the standard A.O.A.C protocols” [16]

Line 148: “Cheese composition was measured at 0, 45 and 90 days of ripening”. The authors did not specify which physico-chemical parameters of the cheese samples were considered. It would be better to report the parameters.

Line 152: the acronym (GC/MS) should be written

Line 153: “(7890-B, Agilent Technologies)”. It would be better to write the name of the city and the nation of the producer.

Line 151:” 2.4.1. CLA Isomers and other Fatty Acids”; the authors must state that the analyses were carried out at 0 days and after 90 days of cheese ripening, as shown in Table 3.

Line 156: “…vortex…”. This word is not correct in this context

Line 160: (Supelco)”. See comment for line 153

Line 163: “L. plantarum “. Then "every sample" would be better understood.

Lines 163-164: A more detailed description of the technique used for colony numbering of L. plantarum should be provided. This will allow authors to use a style that matches that of the Journal “Microrganisms”, (although the authors report the references [19, 20, 21]). Moreover, references can be deleted.

In addition, the following sentence could be prepended and placed at the beginning of the paragraph: “Lp. plantarum was determined after inoculation, pre-acidification, cooking, cheddaring, 30, 60 and 90 days of maturation” . This would help to understand more about the types of samples tested.

Line 165: why did the authors report “….calculated as cfu/ml” and not “calculated as cfu/g of sample”?

Line 171: see comment for line 153

Line 179: “…using standard methods [23]”. The standard methods for the analysis of milk fat composition are reported by Rutten et al. (2009) [23]. In addition, the cholesterol and peroxide values were not determined by these authors. Other authors should be mentioned, also regarding cheese analysis.

Line 183: “…..standard method was followed”. At the end of this sentence a reference should be written.

Lines 184-185: the following sentence is not clear and it should be rewritten: “For the pre-drafting of terminology (color, flavor and  texture), for sittings (each 2 h) were conducted.”

Line 187: “…well-ventilated ….laboratory”. This is not correct as ventilation can affect sample evaluation.

The following ISO norm is suggested: “ISO 8589:2007, Sensory analysis - General guidance for the design of test rooms”.

Lines 188-189: “FIZZ software (Biosystemes, Couternon, France, Version 2.46B)”. It should be better “FIZZ sensory software (version 2.47B, Biosystèmes, Courtenon, France)”.

Line 190: “…distilled water…”

Line 190: reference [24]; the article by Khalid et al. (2017) is not related to sensory evaluation. Authors should include a relevant reference.

Line 191: Authors should indicate the meaning of the abbreviation "IFT".

Line 194:  “….Completely randomized design (CRD)….”

Line 194:” …..the impact  of treatments.” This sentence must be rewritten; it is not complete and explanatory.

From line 198 to line 201: “Free fatty acids (FFA) (0.14%), moisture content (0.19%), peroxide value (POV) (0.25 198 MeqO2/kg), iodine value (141.5cg/100g), unsaponifiable matter (1.34%), saponification value (191.7mg KOH/g) and refractive index were 1.4745 were found in cold extracted safflower oil”.

Two comments need to be borne in mind :1) It is recommended to enter the data in a table.

2) Lines 144-145 report that cold extracted safflower oil was analysed only for FFA, peroxide value (POV), iodine value, saponification value and unsaponifiable matter. Therefore, authors should complete paragraph 2.3.

Line 196: why was the significance level not set?

Line 204: Milesi et al. [25] did not analyse cold extracted safflower oil. Authors should include a relevant reference.

Line 207:”…. to standard cheddar cheese”. There should be a reference at the end of this sentence.

Lines 208-209: authors state that “three strains of starter cultures did not affect the chemical composition of all dry-salted cheeses over 4 months of ripening….”. This statement is only true to a certain extent.

Das et al. [26] used four starters: Lb. fermentum, Lb. rhamnosus, Geotrichum candidum, and Yarrowia lipolytica. Authors should change the sentence.

Furthermore pH, fat moisture salt, and calcium were measured in samples. "Chemical composition" is a generic indication; Authors should indicate physical and chemical parameters determined by Das et al. [26].

Line 209: [26] should be written at the end of the sentence (after “…of ripening”)

Lines 209-210: “…non-starter bacteria did not affect the chemical composition of cheddar cheeses [27]”.

Law et al.(1976) [27] measured only flavour scores, free fatty acids, methyl ketones, and H2S in cheddar cheese samples. The authors would be well advised to include a relevant and more recent reference.

Line 212:” Chemical composition of cheddar cheese……”. Murtaza et al. (2022) [28] analysed cheese samples to also determine pH values and the percentage of protein, moisture, and fat. Authors should report these parameters by replacing the generic phrase "Chemical composition" with the specific physicochemical parameters mentioned above.

Line 214: it is necessary to write [28]at the end of the sentence, after the word “cheese”.

From line 214 to line 216: Khan et al. (2019) [29], in their review, reported the following reference “Ullah, R., Nadeem, M. and Imran, M. (2017)…….Lipids Health Dis., 16(1): 34”. Ullah et al (2017) [not Khan et al. (2019)] studied the gross composition of cheese added with chia oil. Authors must change the reference.

Lines 216-218: Ahmed et al.(2015) evaluated the anti-inflammatory and acute toxicity effects of Baccharoides schimperi (DC.) in order to get new anti-inflammatory agents of natural origin. The article does not have no relation with Gouda cheese.

Lines 218-220:  it is advisable to read carefully the content of the article by Ong and Shah (2009) and to modify the statement that is reported in the text.

Line 221: Table 1 does not show the “Effect of Lp. plantarum and safflower oil on the composition of cheddar cheese”; in this Table the authors report the mean values (%) of moisture, fat, protein, and pH of the Control, T1, T2, T3 and T4. It is recommended to change the caption of Table 1.

For a more complete evaluation of the samples, moisture, fat, protein, and pH should have been combined with additional physico-chemical indices.

From line 224 to line 227: authors have described the composition of “control”, T1-T2, T3 and T4 (in lines 132-139). There is no need to repeat what has already been written.

Line 222: the sentence is not clear.

Line 229-230: “Survival of Lp. plantarum”. The authors studied changes in bacterial loads rather than survival. For this reason, table caption and the beginning of the paragraph should be changed. The same applies to the lines 232, 237 and the caption of Table 2.

From Line 232 to line 237: Further studies are needed to confirm the findings reported by the authors.

Line 240-242: The statements in these lines do not refer to the study by Das et al (2005) [26]. The authors need to correct the reference.

Lines 243-244: Abd El-Salam [32] used Yoghurt starter (Lb. delbreukii subsp bulgaricus and Streptococcus thermophilus, 1:1); Probiotic Lactobacillus casei and Lactobacillus acidophilus were tested for their ability to produce CLA from linoleic acid. These authors stated that “cheese from the different treatments employed retained high counts of the different bacterial groups and strains counted throughout the storage period”.

Lines 244-247: The authors should carefully check the article by Gómez-Torres et al. (2014) [33]. Indeed, their claims do not refer to this article.

Line 247-250: The article at reference [34]is entitledSerum Leptin and Vascular Risk Factors in Obstructive Sleep Apnea”. The authors should write the correct reference.

Line 251: Table 2:” Survival of Lp. plantarum (log CFU g-1) in ripening phase”. In this table only the counts of Lp. plantarum in T1, T2, T3 and T4 are shown. It is recommended to change the caption of Table 2

In addition, the Control samples were not examined. Therefore, it is not possible to check the differences between the control samples, and the T1, T2, T3 and T4 samples.

Line 252: From a technical point of view, the English in this sentence should be improved.

From line 264 to Line 287: the results are showed in Table 3. They need not be repeated. Table 3 could be followed by considerations from [35, 36] and statistical results.

Line 270-271: “Conversion of linoleic acid to CLA was largely dependent upon the concentration of linoleic acid in the  substrate [35, 36]”. Authors are asked: where in articles [35, 36] is such a consideration found?

From line 288 to line 291: The statements in these lines do not refer to the study by Chung et al (2008) [37].The authors need to correct the reference.

Line 292: Phillips et al. [33] are not reported in the References

Lines 296-297: Authors are asked: where in article [38] the considerations are found?

Lines 298-300: Authors are asked: where in article [39] the considerations are found?

Lines 300-302:authors are asked: where in article [40] the considerations are found? Furthermore, reference [40] is the same as reference [13].

Lines 302-304: Collins et al. [41] describe the Cheddar flavour containing mineral oil. authors are asked: where in article [41] is their consideration found?

Lines 304-306: Authors are asked: where in article [30] the considerations are found?

Lines 312-315: Authors are asked: where in article [42] the considerations are found?

Line 327:  The results after 60 days of analysis can be reported in Table 4

Line 316 -327: The authors report the results of statistical analysis; other considerations would have been written.

Lines 335-338: Authors are asked: where in article [43] the considerations are found?

Line 339:”… and microbial activities lead to the production of acetic acid from lactose and citrate”. Authors are asked: where in article [44] is such a consideration found?

Lines 340-341: “Heterofermentative lactobacilli can also produce citric acid from lactose and amino acid”. Where in article [44] is such a consideration found?

Line 345: reference [45]. This reference is reported as follows:” McSweeney, P.L.; Fox, P.F. Advan. Dairy Chem. Springer: 2003; Volume 1”. According to the "Instructions to Authors", this reference is written incorrectly. Therefore, the statements in lines 341 to 345 cannot be verified.

Lines 345-349: “Khan et al. [29] produced……traditional culture”. Where in article [29] are these considerations found?

Line 350:” Table 4. Effect of Lp. plantarum and safflower oil on the organic acids of cheddar cheese (ppm)”.

Table 4 shows the amount (ppm) of organic acids of Cheddar cheese at 0, 45, and 90 days of ripening. It does not show the “effect of Lp. plantarum and safflower oil on the organic acids. It is advisable to change the caption of Table.

Lines 354-359: The authors reported some general considerations about “Lipolysis”; these considerations are sufficient for an Introduction. Conversely, they must be related to the results of the study.

Furthermore, except for the sentence “….it leads to the liberation of fatty acids from triglyceride molecules”, where in article [46] are the considerations found?

Line 360: reference [47] is entitled “Improvement of the oxidative stability of butter oil by blending with Moringa oleifera oil”. Neither Parmesan nor Swiss cheese are covered by this article.

Line 371: reference [41] is entitled “Lipolysis and free fatty acid catabolism in cheese: a review of current knowledge” and does not include the production of Cheddar cheese.

Line 376: reference [30] does not cover “Cheddar cheese samples having 10% chia oil” and FFA contents

Lines 378- 379: at the end of the sentence there should be a reference.

Lines 384-385:reference [48]; the article reports “auto-oxidation and formation of trans fatty acids in milk” and “heating for a prolonged period (30 min) at relatively low temperature (63 ± 1.0°C) under aerobic conditions seems to contribute to lipid oxidation more than heating at higher temperature for 5 min”. There is no quote on cheese.

Line 385: In which parts of this article do Herzallah et al.(2005) talk about POV?

Lines 386-389: at the end of the sentence there should be a reference. Furthermore, this sentence is not necessary; it is unrelated to the following considerations.

Line 391: at the end of the sentence, there should be a specific reference to the EU legislation in question.

Lines 391-393: where in article [49] are the considerations found? Drake's article 82007) deals with sensory analysis of Dairy Foods.

Line 399: Considering the results of the statistical analysis, the sensory properties of the cheese samples were partially affected by Lp. plantarum and safflower oil at concentrations up to 9%.

Line 402: According to the "Instructions to Authors", this reference is written incorrectly. Therefore, the statements in lines 401 to 402 cannot be verified.

Line 402: Singh et al. [51]. According to the "Instructions to Authors", this reference is written incorrectly.

It is likely that the authors are referring to: “T.K. Singh, M.A. Drake, and K.R. Cadwallader (2003). Comprehensive reviews in food science and food safety, 2:166-189”. In this review the authors state that “different single-strain starter cultures produced different flavors”. In “Conclusions” they do not report that “stains of starter cultures can develop pleasant flavor in cheddar cheese”.

Conclusions: “To prevent metabolic disease, CLA should be taken on regular basis.” Many studies are needed before this claim can be made. It is also based on medical evidence.

REFERENCES

-          Authors should check the reference, as several numbers have been given incorrectly and need to be deleted.

-          More recent References should have included

-          Some references do not follow the "Instructions for Authors"

It would be necessary for the authors to check the content of the articles cited in the text.

Extensive editing of English language required

Author Response

Reviewer 3 Report (New Reviewer)

Dear Editor and Authors,

I send you my review about the article “Effect of Lactiplantibacillus plantarum on the Conversion of Linoleic Acid of Vegetable Oil to Conjugated Linoleic Acid, Lipolysis and Sensory Properties of Cheddar Cheese”.

The scope of the paper, as reported in the aim was to use of safflower oil as a source of linoleic acid and Lp. plantarum as microbiological converter of linoleic acid to conjugated linoleic acid in Cheddar cheese.

In my opinion, the paper result well structured and original, however, it need of some little change that I report below.

The introduction is well written and adequately to the aim of the research. However, in this chapter it should report some research that have studied similar aspects in microbiological converter of linoleic acid in Cheddar cheese or in other products similar to it.

The chapter Materials and methods result well structured and complete. Nevertheless, the title of the sub-paragraph, like for example the 2.6 one should be separated from the top row

The results is very well presented and they are very well discussed, also in comparison to the data reported in the literature. However,

the tables format should be checked in relation to the Authors guideline.

Moreover, table 4 need to be placed in a single page.

Finally, the conclusions of the paper result adequate to the results showed and they satisfy the aim of the research.

Nevertheless, the section of the conclusions should not be limited only to reporting a summary of the data already reported, but should also include some personal comments of the Authors.

In this regard, I would suggest that the authors report their opinion on the impact that the results of their paper could have.

Best regards

Author Response

This manuscript is a resubmission of an earlier submission. The following is a list of the peer review reports and author responses from that submission.

Round 1

Reviewer 1 Report

The manuscript  of Khan et al. describes the production of cheedar cheese enriched in CLA by means of a strain  of Lactiplantibacillus plantarum and different proportions of safflower oil. Although they obtain interesting results, the manuscript has seriuos drawbacks that need to be solved in order to be acceptable for publication.

First of all, the manuscript is poorly written and the results are hardly discussed. English should be checked throughout the manuscript.

The methods are poorly described and also some of the references they mention have nothing to do with them, e.g, 15, 17 and 18. Methods should be described with more detail and appropiated references added.

The “Statistical Analysis” section seems to be copied  from another work.

The “Results” section should be “Results and Discussion”. In any case, the results are poorly described. The data shown in the tables are repeated in the text, making it difficult to follow. And the discussion of results is poor and poorly written.

 The reference list is nonsense. Some of the articles they include have nothing to do with the study, e.g., 7, 9, 21, 34, 35...Reference 10 is not mentioned in the text; 15 and 16 are practically the same…

This fact, together with others mentioned, indicates that the authors barely reviewed the manuscript before sending it.

More especific comments

Abstract

Line 31: ..As CLA is not a essential nutrient, authors cannot say “the dietary requirements” or the “need” of CLA, but “the recommendation” . Please, revise this all along the manuscript.

Line 35: Please, especify what 85% means:  85 g LA/ 100 g oil?  85 g /100 g total FA? Please, clarify this all along the manuscript.

Lines 45-46: In y opinion, the statement is not correct. In table 5 differences in FFA and POV can be seen in 90 days ripened cheeses. These differences are important as cheese is consumed at this point of ripening.

Abbreviations

CRD is described twice.

LA is used for two different compunds (linoleic acid and lactic acid). Please, correct it all along the manuscript.

In my opinion, chemical formulas can not be considered as abbreviations.

Introduction

Lines 79-80: Include some references to support the statement.

Line 88: “processes” instead of “properties”

Lines 89-90: The reference [11] does not describe what  is said in the statement.

Lines 106-107: ….processes, as lactose metabolism, proteolysis, lipolysis and production of free fatty acids (FFA) and flavoring compounds production [13]. FFA composition…

Lines 108-109:    . CLA may be oxidized during ripening in enriched cheese..

Materials and Methods

As I said before, in my opinion, methods should be described with more detail and the references should be corrected as many of them do not describe the mentioned method.

In “Exerimental Plan” authors should explain how many litters of milk do they used for each cheese batch, how many cheeses (kg) where obtained... Do all types of cheese have the same fat content? I mean, is the milk fat replaced by safflower oil to get the same fat concentration? Or is safflower oil added to the milk fat and consequently cheeses made with added oil have higher fat content?

What means “In triplicate”? Do they analyze three cheese for each type of cheese? Or the same cheese was analyzed three times?

In section 2.4.1. CLA Isomers and other Fatty Acids

The specific analysis of CLA isomers is very poorly described. In addition, they must explain how they do the quantification and in what units the content of FA is described.

In section 2.4.2. Population of Lactiplantibacillus plantarum

MRS agar is a medium for the cultivation and enumeration of all Lactobacillus spp. How they count especifically Lactiplantibacillus plantarum?

2.4.4. Lipolysis

Cholesterol concentration and peroxide value do not describe the level of lipolysis.

Results

It should be “Result and Discussion”

As I said before, the data showed in the tables should not be repeated in the text, in general. They can be summarized and the most important results underlined and discussed.

Lines 189-192: The data described are not in table 1. In any case, I do not see their interest in this work. They can be added as supplementary data.

Line 249:   …of LA enriched cheddar cheese…( the results of CLA have not yet been described).

Lines 253-264:  All the ideas described in the paragraph has been already said in introduction.

Line 318: mg FA/ 100g cheese?

Lines 372-373:  I do not see the need to include the data of other cheeses  if they cannot be compared with those obtaines in the present study.

Line 377: At 90 days ripening table 5 shows that T1 has higher FFA than control.

Lines 383-388. What is the reason for these results?

Higher especificity of microbial lipases for unsaturated FA present in this type of oils?

Lines 389-395: Lipolysis is the hydrolisis of TG and production of FFA. So, there is no sense in relating it with the cholesterol content. On the other hand, the only reason that could explain the reduction of cholesterol cocentration is a effect of dilution because of the addition of safflower, because, a far as I know, very few strains of bacteria have the ability to degrade this lipid.

Lines 401-405:  They do not mention differences in peroxide value between cheese types at 90 days of ripening that, in my opinion, are worth mentioning.

Conclusion

Line 423: They concluded that the recommendation 2g CLA/day can be achieved from one slice of cheese. In relation to that, first of all, not all CLA isomers have the same effect on health. Some of them have no effect and others can be harmful. They have not discuss about it at all.  cis9,trans11—CLA is the one related to favorable effects. The highest concentration of this isomer in cheese is of 1.91 mg/ 100 g of cheese (in 90 days T4 ripened cheese). Therefore, according to this, it would be necessary to eat 100 kg of cheese every day...

Line 424: I do not agree with the statement that lipid oxidation of CLA enriched cheddar cheese is not different from standard cheddar cheese.

Some other minor comment are added in the revised manuscript

Some of the issues that I mention to the authors seem quite serious to me, such as the fact that the reference list has multiple errors. This indicates that the authors have not adequately reviewed the manuscript before submitting it, which represents an added effort for the reviewers. And this, as I comment, seems to me a serious failure on the part of the authors.

Reviewer 2 Report

The authors present an interesting, innovative manuscript. The conclusions are particularly well written. However, three main issues must be solved; the most important of which regards the sensorial experiments presented in the manuscript, because of its ethical nature. According to the IFT, sensorial experiments are to be regarded as experiments with humans, and need, therefore, the signature of a form of informed consent by the participants. The authors need to clarify whether this procedure was followed, as well as any others that, by the regulations in their institutions and the law in their countries, are applicable to experiments on human subjects. Another important issue is the assumption that the authors use mixed starter cultures, in which Lp. plantarum, the bacterium they are assessing for CLA production under cheesemaking conditions, is just one of the lactic acid bacteria (LAB) added to the cheese, can be counted on MRS Agar. This culture medium is not selective, and other LAB, both from the added starter and of adventitious nature, can grow on MRS Agar. Therefore, this part of the discussion must be rewritten. Lastly, the statistical analysis methodology used is not adequate for the data obtained in sensorial testing, which are non-continuous variables. Accordingly, statistical analysis for this part of the manuscript must be redone, using appropriate methods.

Comments to several other details are given below.

Lines 73-74 – Adding a concise sentence about the role of pasture-based diets on CLA concentrations in milk and in dairy products would be interesting for the potential readers of this paper. In this sentence, the authors could also include information on the impacts of seasonality on CLA content in products derived from pasture-fed animals.

Line 87 – Please correct “Lp. Plantarum” to “Lp. plantarum" (also on line 243).

Line 157 – “… 30, 60 and 90 days” – Do these refer to numbers of maturation days? If so, please state “… 30, 60 and 90 days of maturation”.

Lines 123 – 135 – Bacterial growth, in general, and that of Lp. plantarum in particular, in food matrices such as Cheddar cheese, is influenced by numerous parameters, among which temperature, nutrient and water availability. The different temperature profiles and unit operations applied during the transformation of milk in cheese influence the interpretation a reader will have of your data. Therefore, cheesemaking needs to be better detailed in this section. Please describe the relevant steps in the preparation of cheese, including times, temperatures and, if available, pH and moisture/water activity. Also include the amount of salt added (or NaCl concentration values). What was the weight of each experimental cheese? What were the milk pasteurization parameters (temperature, time)? You can present this information under the form of text or under the form of a process flow diagram (flow chart) containing each of the cheese manufacture steps and the respective parameters, as above mentioned, to keep the manuscript concise, but it should be presented in more detail than that given by a mere citation.

Line 156 – “Number of Lp. plantarum…” Please keep in mind that MRS is not a selective medium – it is, at best, elective. You cannot be sure that the counted colonies belonged to Lp. plantarum and not to other starter culture constituents – or even to non-starter lactic acid bacteria. Therefore, you should refer MRS counts not as “Number of Lp. plantarum”, but rather as “Numbers of lactic acid bacteria”.

Please correct the text accordingly.

Line 132 – “… crisp bread and distilled were provided…” Do you mean “…crisp bread and distilled water were provided…”? Please correct accordingly.

Lines 174 – 183 – According to the IFT, sensorial testing is regarded as experimentation on human subjects. Please make sure that all applicable regulations were followed, namely, the signature of an informed consent form and previous approval by the Committee for Ethics of the authors’ institutions, if required in their countries/institutions. Information regarding this aspect must be included in your manuscript, namely the number/reference of the permit granted by the Ethics Committee and the indication that an informed consent was signed by all participants in the test.

Line 250 – Do you mean rows or columns? It would make more sense if it were columns. Please correct, if applicable.

Lines 218 – 250 – As discussed above, regarding the (very low) selectivity of MRS Agar, the authors cannot infer that they assessed survival of Lp. plantarum with counts on this culture medium. What was obtained was merely a total lactic acid bacteria count (LAB) – which would include not only the members of the former Lactobacillus genus (such as Lp. plantarum), but also any other members of the starter culture, as well as non-starter (adventitious) LAB. Therefore, this whole section needs to be rewritten, bearing in mind that the information given by MRS counts is a total LAB count, not specifically a count of the added Lp. plantarum. The L. lactis strains added as part of the starter culture can also yield colonies on MRS agar, and so do the non-added, adventitious, non-starter LAB that are part of the naturally present microbiota of most cheese varieties, even in those that were made from pasteurized milk. Enterococci, for instance, are thermoduric, will survive pasteurization, are part of the non-starter LAB and will grow on MRS Agar. So, please revise your text according with this information.

Table 6 – The statistical analysis used is not adequate for this type of variables (non-continuous variables). Please use adequate statistical methodologies for this type of data.

Lines 419 – 427 – This part is well written and it was a great idea to show that the novel process you propose would allow consumers to fulfill their daily needs of CLA without exceeding their daily allowance of dairy products.

Some minor faults were detected and should be corrected in the next version(s) of the manuscript.